# Design, Cost Estimation and Sensitivity Analysis for a Production Process of Activated Carbon from Waste Nutshells by Physical Activation

**Marcelo León [1,\*], Javier Silva [1] , Samuel Carrasco [1] and Nelson Barrientos [2]**

[1] Escuela de Ingeniería Química, Pontificia Universidad Católica de Valparaíso, Av. Brasil 2162, Valparaíso 2362854, Chile; javier.silva@pucv.cl (J.S.); samuel.carrasco@pucv.cl (S.C.)

[2] Trazado Nuclear e Ingeniería Ltda., Francisco de Villagra 385, Santiago 7760016, Chile; n.barrientos@trazadonuclear.cl

[\*] Correspondence: marcelo.leon@pucv.cl; Tel.: +56-32-237-2630

**Abstract:** A conceptual design of an industrial production plant for activated carbon was developed to process 31.25 tons/day of industrial waste nutshells as the raw material and produce 6.6 ton/day of activated carbon using steam as an activation agent. The design considered the cost of the main equipment, the purchase price of the nutshells, basic services, and operation. A sensitivity analysis was developed, considering the price of the finished product and the volume of raw material processing varied up to ±25%. Furthermore, the total annual cost of the product was determined based on the production of 2100 tons/year of activated carbon. Two cash flows were developed and projected to periods of 10 years and 15 years of production, using a tax rate of 27%, a low discount rate (LDR) of 10% per year, and without external financing. For a 10-year production project, the net present value (NPV) was USD 2,785,624, the internal return rate (IRR) 21%, the return on investment (ROI) 25%, and the discounted payback period (DPP) after the fifth year. Considering a project with 15 years of production, the NPV was USD 4,519,482, the IRR at 23%, the ROI 24%, and the DPP after the fifth year of production.

**Keywords:** economic evaluation; production cost; nutshell waste; activated carbon

## 1. Introduction

Currently, there is a great interest in adopting more efficient and low-cost processes for the treatment of wastewater. The rapid growth of the world population is resulting in increased contamination of freshwater sources, generating conditions of water stress in the short term [1]. The adsorption technique using activated carbon is one of the most-used methods for the removal of organic pollutants and metal ions in wastewaters, with previous studies reporting effective removal of impurities [2,3].

Activated carbons are highly porous carbon materials with a high specific internal surface area and commonly serve as adsorbent material in various industrial separation and purification applications [4]. Activated carbons can be obtained from chemical activation processes or physical activation of organic precursors. Chemical activation consists of the impregnation of the raw material with chemical agents, such as KOH, $ZnCl_2$, and $H_3PO_4$, among others, and simultaneous carbonization (pyrolysis) of the impregnated biomass in an inert gaseous atmosphere, where the main role of the activating compound is the degradation of the cellulosic material [5,6]. On the other hand, two relevant processes are involved in physical activation. The first process involves the pyrolysis of the raw material in an inert atmosphere that usually uses a gaseous stream of nitrogen. The second stage of activation involves gases, such as water vapor, carbon dioxide, or a mixture of these gasses with nitrogen or air in different proportions at high temperatures for the final activation [5,6].

Generally, the raw material's pyrolysis process takes place at temperatures between 400 and 600 °C, while the activation stage takes place between 800 and 1100 °C [5,6]. In comparison with chemical activation, physical activation can be considered clean and environmentally friendly, which would avoid the incorporation of impurities from chemical activation agents. There are various organic materials that serve as raw materials for obtaining activated coals, such as coconut shells [7], rice shells [8], palm shells [9], peanut shells [10], and nutshells [11], among others [5,6]. Recent studies have shown that physically activating carbon with steam using nutshells as a precursor has presented interesting characteristics for applications in the removal of heavy metals from effluents of polluted waters [12,13]. In this sense, it could be interesting to develop a complete economic evaluation of a nutshell activated carbon production plant, using the parameters and conditions used in the laboratory.

There are few reports on the analysis of production costs of activated carbon production plants. Noticeably, the study of Ng et al. [14] in 2003 considers the production-cost analysis for an activated carbon plant from pecan shells. The study compares the physical activation with steam and the chemical activation with phosphoric acid and reaches a production cost of 2.72 USD/kg and 2.89 USD/kg, respectively. Choy et al. [15] in 2005 reported on the production of activated carbon from bamboo waste and the evaluation of two production plants: one independent and one integrated. The study presents a thorough economic investigation and sensitivity analysis, estimating values for the internal return rate of 13.0% and 20.1%, respectively, among other economic indicators of interest. Subsequently, Lima et al. [16] in 2008 reported the capital and operating costs for an activated carbon plant from poultry waste, obtaining a global production cost of 1.44 USD/kg. Next, Stavropoulos et al. [17] in 2009 reported different production costs and other economic indicators for physically and chemically activated carbon production processes based on precursors, such as used tires, wood, petroleum coke, carbon black, coal, and lignite. The authors obtained production costs of 2.23 USD/kg, 2.49 USD/kg, 1.08 USD/kg, 1.22 USD/kg, 1.25 USD/kg and 2.18 USD/kg, respectively. Moreover, Vanreppelen et al. [18] in 2011 reported results on the feasibility of a process to produce nitrogenous activated carbon by co-pyrolysis of a mixture of particle board (chipboard) and melamine formaldehyde resin, estimating different economic indicators and developing sensitivity analyses. Furthermore, Santadkha and Skolpap [19] in 2017 reported the results of the economic evaluation for three types of production processes: first, a process of generating activated carbon from coconut shells; second, a process of regenerating coal obtained from the petrochemical industry, and, third, an integrated process that combines the production and regeneration of activated carbon. Nandiyanto [20] in 2018 reported the economic feasibility of the production of activated carbon and silica particles from rice straw residues, obtaining various economic indicators, such as an internal return rate of 44.06% for a case study.

The present work updates, complements, and discusses new economic approaches on the implementation of an industrial plant that produces physically activated carbon from nutshells. Results can be extrapolated to various types of raw material, as long as the selected raw materials and process conditions are similar. Although the assumptions made for the selected production process provide an adequate initial technical basis for the economic evaluation of a production plant at the industrial scale, there are limitations related to the conditions provided in this particular work. In this sense, the precision of this study is highly dependent on the variation in the cost of the main equipment, the cost of the raw material, the sale price of the product, and the estimated operating conditions of the production process. Another weakness may be related to the use of the factorial method to estimate the total capital investment used to calculate the economic indicators of profitability. Future work should be directed towards the optimization of the selected production process, through the use of a suitable chemical process simulation software. This information would greatly improve the economic study.

## 2. Materials and Methods

### 2.1. Production Process

The production process and operational parameters were proposed based on recent bibliographical data, which reported the obtaining of activated carbons from nutshells, using an inert nitrogen-based atmosphere for the pyrolysis stage and a flow of steam as an activating agent for the activation stage [13]. The parameters and conditions reported by Nazem et al. [13] were considered for the conceptual plant design considering a specific area of 1248 $m^2$/g for the activated carbon. This report indicates the pyrolysis stage temperature of 600 °C and a residence time of 1 h. For the activation stage, the operational conditions considered were 950 °C for temperature and 1 h for residence time. In the present work, a conversion rate of 30% was considered for the pyrolysis stage, in which nutshells are transformed into natural carbon, and a conversion rate of 70% for the activation stage, in which previously obtained natural carbon is converted into activated carbon. An overall conversion rate of 21% was established; consequently, this percentage of the available raw material of nutshells is finally transformed into activated carbon. On the other hand, material and energy balances were developed to obtain all the input and output flows of the production process, adjusted to the operating conditions considering production for the base case of 6.6 ton/day of activated carbon with ±25% variation. A sufficiently wide range of variation was considered to adequately establish the effect of the parameters on the net present value. Finally, the requirements for basic water, nitrogen, steam, and fuel services necessary for the operation of the process were estimated, as well as the size parameters for the sizing of the main process equipment.

### 2.2. Economic Analysis

For the economic evaluation, the installation of the production process plant in Chile was considered, because it has an appropriate availability of the raw material, access to ports of shipment, stability, and economic integration at a global level. On the other hand, considering that the mass ratio between the internal fruit and the shell is close to one, it was possible to establish that the quantity of available nutshells in Chile is similar to the exports of shelled nuts considered under the International Trade Center (ITC) code 080232. In this way, it was estimated that the availability of raw material was close to 34 kton/year, sufficient to supply the 10 kton/year (±25%) needed for the proper development of the investment project.

A working year of 360 days was considered, with a typical working period of 320 days/year for production and 40 days/year for plant maintenance tasks, and with a density of 600 kg/$m^3$ in the case of nutshells and 500 kg/$m^3$ in the case of the activated carbon obtained to size the main process equipment. The cost of the main equipment was estimated from actual local commercial quotations and costs present in the available literature [14–16]. The costs were updated to present value using the chemical engineering plant cost index (CEPCI), applying Equation (1). The costs of the main equipment were adjusted for required production capacity through Equation (2), using a typical exponent of 0.6 (six-tenths-factor rule) [21,22]. The operational labor costs were estimated graphically for an average condition plant, yielding 27 employee-hours/day/processing step.

$$\text{Cost Item (2019)} = \text{Cost Item (20XX)} \times \left[ \frac{\text{Cost Index 2019}}{\text{Cost Index 20XX}} \right] \tag{1}$$

$$\text{Cost New Capacity} = \text{Cost Old Capacity} \times \left[ \frac{\text{New Capacity}}{\text{Old Capacity}} \right]^{0.6} \tag{2}$$

The total capital investment established from the total cost of the main process equipment installed at the plant—including the auxiliary equipment, the total product cost including the operational labor, the projected cash flows using the selling price of the product and the sensitivity analysis—were determined taking as a guide the standard procedures described in *Plant Design and Economics for*

*Chemical Engineers* by Peter & Timmerhaus [21,22]. In particular, the total capital investment was based on the percentage of the delivered equipment cost method for a solids and liquids processing plant, described in the previous reference. The depreciation of process equipment was calculated using the linear method, and the income tax rate used was 27% (case of Chile) [23]. The expected error in the estimation of factored costs is around ±25%. The sales price of the finished product (activated carbon) for the base case was estimated at 2.75 USD/kg. Finally, economic indicators, such as net present value (NPV), internal return rate (IRR), return on investment (ROI), and discounted payback period (DPP), were determined. The equations for calculating NPV (Equation (3)), IRR (Equation (4)), and ROI (Equation (5)) are shown below [15,17]. In the case of the DPP, it was estimated graphically using the discounted cumulative cash flow.

$$\text{NPV [USD]} = \sum_{t=1}^{N} \frac{F_t}{(1+d)^t} - I_0 \tag{3}$$

$$\text{NPV} = \sum_{t=1}^{N} \frac{F_t}{(1+d^*)^t} - I_0 = 0 \rightarrow d^* = \text{IRR} \tag{4}$$

$$\text{ROI [\%]} = \frac{F_t}{I_0} \times 100 \tag{5}$$

where N = project duration [years], $F_t$ = annual profit [USD], d = low discount rate (LDR = 10% average market value), $I_0$ = total capital investment [USD], and d* = internal return rate (it is the discount rate when NPV = 0).

## 3. Results

### 3.1. Production Process Description (Base Case)

A processing capacity of 31,250 kg/day of nutshells was considered for the production of 6563 kg/day of activated carbon from nutshells for 320 effective working days of operation. Figure 1 shows the proposed production process for obtaining activated carbon from nutshells. The first stage is to grind 31,250 kg/day of nutshells to reach a maximum granular particle size of 18 mesh (1.0 mm) in a mill capable of processing 1302 kg/h and then to collect the ground raw material in two storage silos with a capacity of 182 m³ each, allowing the raw material stock to be maintained for one week of production.

Subsequently, in the second stage, the 31,250 kg/day of ground nutshells must be processed in six independent loading processes per day of 5208 kg/load. The nutshells are fed to a rotary kiln where the raw material's carbonization process is carried out at a temperature of 600 °C with a residence time of 1 h. To avoid the combustion of organic matter inside the furnace chamber, a nitrogen flow is incorporated to displace the air. In this way, 1563 kg/load of natural coal is obtained from nutshells as an intermediate product, with an estimated yield for this stage of 30%. The process of activation is carried out at a temperature of 900 °C in the presence of a steam current as an activating agent for a residence time of 1 h. 1094 kg/load of activated carbon from nutshells is obtained as the final product of this stage, with a yield of 70%. The rotary reactor was designed with a volumetric load ratio of 15% and an internal volume of approximately 58 m³. In the third stage, the activated carbon produced passes to cooling equipment where the temperature decreases to 50 °C using a nitrogen atmosphere and is stored in a silo of 92 m³ of capacity that means the stock can be kept for a week of production. The fourth stage consists of passing the activated carbon through sieving equipment capable of processing 273 kg/h of product. Finally, in the fifth stage, the packaging process of the activated carbon produced is developed.

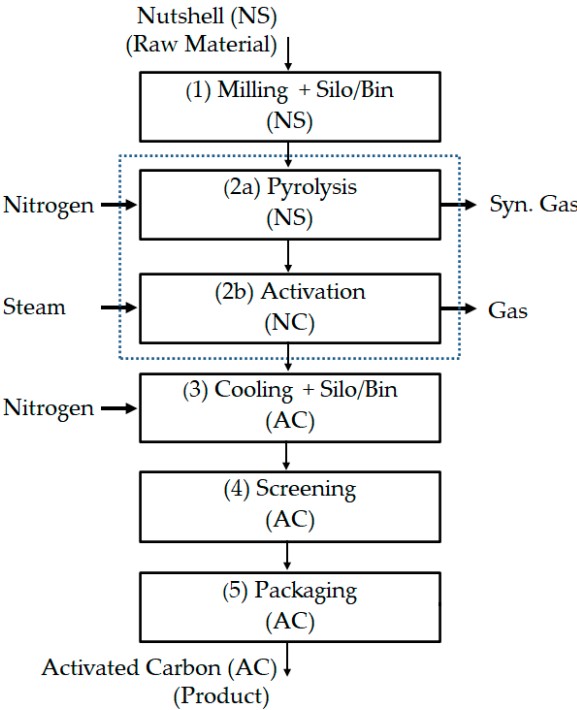

**Figure 1.** The proposed production process for obtaining activated carbon from industrial waste nutshells by physical activation with steam.

### 3.2. Cost Estimation (Base Case)

The cost of the main equipment of the production process placed in the plant for the base case considered is shown in Table 1. The total cost of the equipment, including auxiliary equipment, is USD 1,063,044. Furthermore, it was considered that the cost of the auxiliary equipment is 10% of the main process equipment. The construction material of the equipment considered was carbon steel.

**Table 1.** Main process equipment specifications and costs for the base case.

|   | MAIN PROCESS EQUIPMENTS | CAPACITY | UNITS | [USD] |
|---|---|---|---|---|
| 1 | Mill (Nutshell) | 1302 | kg/h | 22,245 |
| 2 | Silo/Bin (Nutshell Storage) | 292 | $m^3$ | 103,473 |
| 3 | Rotary Kiln Pyrolysis/Activation (6 process per day) | 5208 | kg/process | 515,624 |
| 4 | Rotary Cooler Activated Carbon (6 process per day) | 1094 | kg/process | 78,479 |
| 5 | Silo/Bin (Activated Carbon Storage) | 84 | $m^3$ | 34,297 |
| 6 | Vibrating Screen Sieve (Activated Carbon) | 273 | kg/h | 7546 |
| 7 | Packaging (Activated Carbon) | 273 | kg/h | 45,816 |
| 8 | Diesel Boiler | 563 | kg/h | 27,896 |
| 9 | Fuel Diesel Tank | 109 | $m^3$ | 68,488 |
| 10 | Cooling Water Tank | 174 | $m^3$ | 62,540 |
| C1 | TOTAL MAIN EQUIPMENT COST | | | 966,404 |
| C2 | TOTAL MAIN + AUXILIARY EQUIPMENT COST (1.1 × C1) | | | 1,063,044 |

By using the total cost of the process equipment (including auxiliary equipment) as the 100% value, the total capital investment for the base case was estimated at USD 5,427,905, as shown in Table 2. In this sense, the total fixed-capital investment reached a value of USD 4,613,613 and a working capital value of USD 814,292. Concerning the base case, the annual cost of raw materials reached a value of USD 1,124,894 per year. This amount considered the purchase of 10 kton/year of nutshells at a unit price of 100 USD/ton, the supply of boiler water of 2700 $m^3$/year at a unit service price of 4 USD/$m^3$, and the supply of liquid nitrogen of 185 ton/year at a unit price of 618 USD/ton. On the other hand, the cost for basic fuel and cooling water services reached a value of USD 936,936 per

year, considering a diesel oil supply of 1162 m$^3$/year at a unit price of 800 USD/m$^3$ and the supply of cooling water of 1866 m$^3$/year at a service price of 4 USD/m$^3$. Operational labor costs were estimated at USD 280,800, considering 27 employee-hours/day/processing step, five process stages as described in Figure 1, 320 days of annual operation, and a cost of 6.5 USD/employee-hours. All mentioned costs were estimated based on local values, considering a high rank to ensure an economic evaluation in the most extreme case (the case of Chile). Consequently, the total cost of the product for the base case was estimated at USD 4,523,987, as shown in Table 3.

**Table 2.** Total capital investment for the base case.

| | ITEM | | [USD] |
|---|---|---|---|
| **A** | **TOTAL FIXED-CAPITAL INVESTMENT** | **A1 + A2** | **4,613,613** |
| A1 | TOTAL DIRECT PLANT COST | 1 to 9 | 3,274,177 |
| 1 | Delivered main equipment (includes auxiliary equipment) | 100% | 1,063,044 |
| 2 | Purchased-equipment installation | 39% | 414,587 |
| 3 | Instrumentation and controls (installed) | 26% | 276,392 |
| 4 | Piping (installed) | 31% | 329,544 |
| 5 | Electrical (installed) | 10% | 106,304 |
| 6 | Buildings (including services) | 29% | 308,283 |
| 7 | Yard improvements | 12% | 127,565 |
| 8 | Service facilities (installed) | 55% | 584,674 |
| 9 | Land (purchase is required) | 6% | 63,783 |
| A2 | TOTAL INDIRECT PLANT COST | 10 to 14 | 1,339,436 |
| 10 | Engineering and supervision | 32% | 340,174 |
| 11 | Construction expenses | 34% | 361,435 |
| 12 | Legal expenses | 4% | 42,522 |
| 13 | Contractor's fee | 19% | 201,978 |
| 14 | Contingency | 37% | 393,326 |
| **B** | **WORKING CAPITAL** | **15 + 16** | **814,292** |
| 15 | About 15% of total capital investment | 75% | 707,988 |
| 16 | Safety and hazard analyses | 10% | 106,304 |
| | **TOTAL CAPITAL INVESTMENT** | **A + B** | **5,427,905** |

**Table 3.** Total annual product cost for the base case.

| | ITEM | | [USD] |
|---|---|---|---|
| **C** | **MANUFACTURING COST** | **C1 + C2 + C3** | **3,821,529** |
| C1 | DIRECT PRODUCTION COSTS | 1 to 8 | 2,862,319 |
| 1 | Raw materials (calculated) | - | 1,124,894 |
| 2 | Operating labor (calculated) | - | 280,800 |
| 3 | Direct supervisory and clerical labor (17.5% of operating labor) | 17.5% | 49,140 |
| 4 | Utilities (calculated) | - | 936,936 |
| 5 | Maintenance and repairs (6% of fixed-capital investment) | 6.0% | 276,817 |
| 6 | Operating supplies (15% of cost for maintenance and repairs) | 15.0% | 41,523 |
| 7 | Laboratory charges (15% of operating labor) | 15.0% | 42,120 |
| 8 | Patents and royalties (4% of C1.1 to C1.7) | 4.0% | 110,089 |
| C2 | INDIRECT PRODUCTION COSTS | 9 to 11 | 595,156 |
| 9 | Depreciation (10% of fixed-capital investment) | 10.0% | 461,361 |
| 10 | Local taxes (2.5% of fixed-capital investment) | 2.5% | 115,340 |
| 11 | Insurance (0.4% of fixed-capital investment) | 0.4% | 18,454 |
| C3 | PLANT-OVERHEAD COSTS (60% of 2 + 3 + 5) | 60.0% | 364,054 |
| **D** | **GENERAL EXPENSES** | **14 to 16** | **702,458** |
| 14 | Administrative costs (15% of 2 + 3 + 5) | 15.0% | 91,014 |
| 15 | Distribution and selling costs (11% of manufacturing cost) | 11.0% | 420,368 |
| 16 | Research and development costs (5% of manufacturing cost) | 5.0% | 191,076 |
| | **TOTAL PRODUCT COST** | **C + D** | **4,523,987** |

## 4. Discussion

It is important to mention that the costs reported in the bibliography are linked to different time periods, which may affect the comparisons with respect to the values estimated in this work. In this sense, the cost comparison described below considers this as a limitation of the present work. As established in Table 1, the cost estimate of the main equipment and auxiliary equipment reached an updated value of USD 1,063,064, which is, in some cases, similar to the costs found in the literature for activated carbon production processes, using the physical route with steam as an activating agent. This value was estimated considering a plant size to process 6.6 ton/day of waste nutshells as raw materials. On the other hand, some of the most current costs are related to the estimate made by Lima et al. [16] in 2008, which reported a total cost of main equipment of USD 1,776,000, to process 20 tons/day of poultry litter. Subsequently, Stavropoulos et al. [17] in 2009 reported a total cost for process equipment of USD 1,154,416, for a production size of 4.5 ton/day considering various raw materials. More recently, Santadkha and Skolpap [19] in 2017 reported a total cost of main machinery and equipment of USD 1,301,429 for the generation of activated carbon from coconut shells and the regeneration of spent activated carbon obtained from the petrochemical industries, considering a plant capacity of 12 ton/day and 10 ton/day, respectively.

As previously mentioned, Table 3 shows that the total manufacturing cost for the base case reached a value of USD 3,821,529, and the total cost of the product reached a value of USD 4,523,987. If we divide these values by the total annual activated carbon production quantity of 2,100,000 kg/year, we obtain a manufacturing cost of USD 1.82 per kg of product and USD 2.15 per kg of product, respectively. For the calculation of these values, a non-zero cost for the raw material of USD 1,124,894 per year was considered. This makes a difference with other published studies that consider a marginal cost or do not consider costs in this important item, which can decrease the total cost of the product and lead to unrealistic conclusions about production costs, since any waste material used in industrial processes acquires economic value [17].

The values mentioned above can be compared with some costs reported in similar studies, being lower, for example, than the cost of the product obtained by Ng et al. [14] in 2003 who reached a production cost of activated carbon from pecan shells of USD 2.72 per kg and USD 2.89 per kg when the process was carried out by physical activation with steam and chemical activation with phosphoric acid, respectively, considering a minimum cost of USD 35,000 for raw materials. The costs obtained in the present work compare favorably with the costs reported by Stavropoulos et al. [17] in 2009 who reported production costs of USD 2.23 per kg, USD 2.49 per kg, and USD 2.18 per kg for the production of physically activated carbon with steam, using worn tires, wood, and lignite as raw materials, respectively. It is important to highlight that these last values were obtained considering a zero cost for raw materials. On the other hand, in the same study, lower production costs of USD 1.92 per kg, USD 1.84 per kg, and USD 1.72 per kg were reported for the production of chemically activated carbon with KOH, using worn tires, wood, and lignite as raw materials, respectively, considering a zero cost for raw materials. However, when a non-zero value was considered for raw materials, higher costs were calculated as USD 11.4 per kg, USD 6.39 per kg, and USD 5.38 per kg for the same precursors, respectively. In another study developed by Lima et al. [16] a lower production cost of USD 1.44 per kg was reported for an activated carbon plant from poultry waste, using physical activation with steam and a subsequent washing step with hydrochloric acid, followed by a step rinse with water. However, the cost was obtained considering a cost for raw materials of USD 269,537 per year, including the transportation of poultry waste, which is below the value considered here.

### 4.1. Economic Evaluation (10-Year Production Project)

Figure 2 shows the cumulative discounted cash-flow diagrams (CDCF) at different applied discount rates and without external financing, to determine the payback period for the capital investment. The payback period was after the fourth year of production, giving a net present value of USD 7,939,235 for a zero discount rate. For a 10% discount rate, the payback period was after the fifth

year of production, giving a net present value of USD 2,785,624. Finally, for a discount rate of 20%, the investment recovery period was after the ninth year of production, giving a net present value of USD 176,231. In all cases the return on investment was 25%. On the other hand, Figure 3 shows the variation curve of the net present value at different discount rates. The internal rate of return (IRR) was estimated at 21%, which is higher than the minimum discount rate (LDR) of 10%, which is generally used by companies for the evaluation of investment projects in chemical plants; therefore, the project is viable for this particular condition.

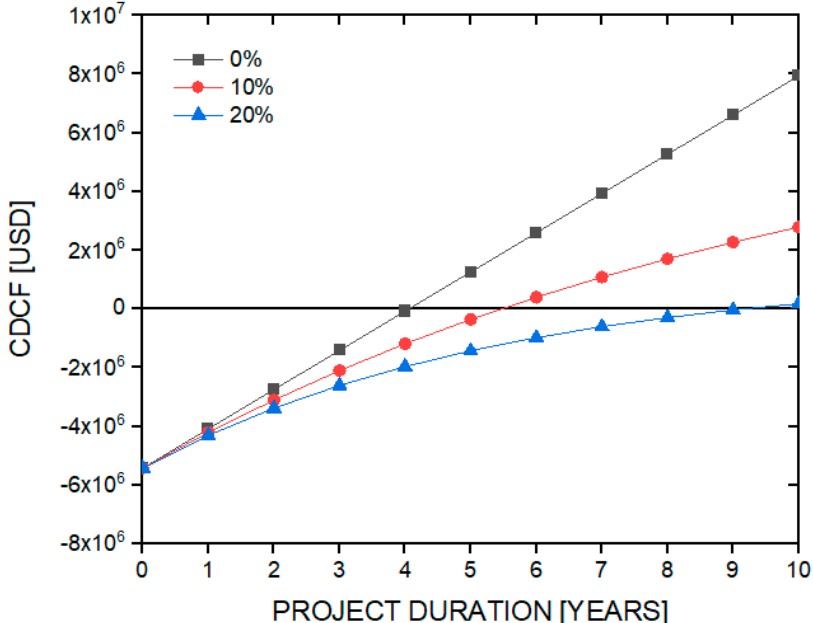

**Figure 2.** Cumulative discounted cash-flow (CDCF) diagrams at different discount rates of 0%, 10%, and 20%, for the 10-year production investment project.

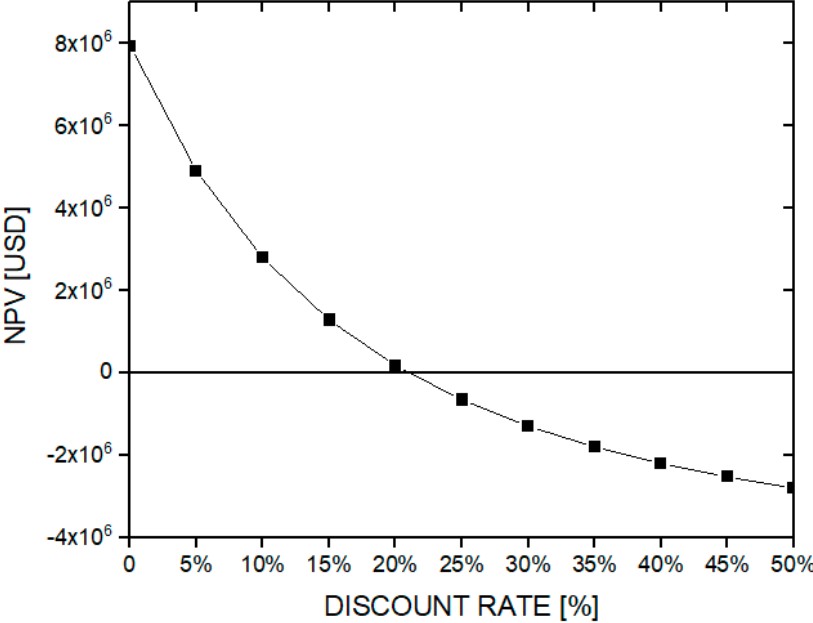

**Figure 3.** Net present value (NPV) of the investment project at different discount rates for the 10-year production investment project.

*4.2. Sensitivity Analysis (Base Case of 10 Years of Production)*

Figure 4 shows the effect on the NPV of the project's cash flow at ten years of production as a base case (using a tax rate of 27% and an LDR of 10% per year) due to the variation of ±25% in the cost of the equipment delivered to the plant, including the auxiliary equipment, the purchase price of the raw material (nutshells), the cost of basic services, and the cost of operational labor. The NPV varied by up to ±78% when the cost of equipment delivered was modified from the base case (USD 1,063,044). When the equipment delivered cost changed by +25%, a minimum NPV of USD 611,708 was obtained, along with an IRR of 12% and an ROI of 18% for this case. On the other hand, the NPV had a maximum variation of ±49% when the nutshells' raw material price changed for the base case (100 USD/ton), this variation being lower than the previous example. The NPV reached a minimum value of USD 1,432,787, when the price of the raw material was increased by 25%, obtaining an IRR of 16% and an ROI of 21%.

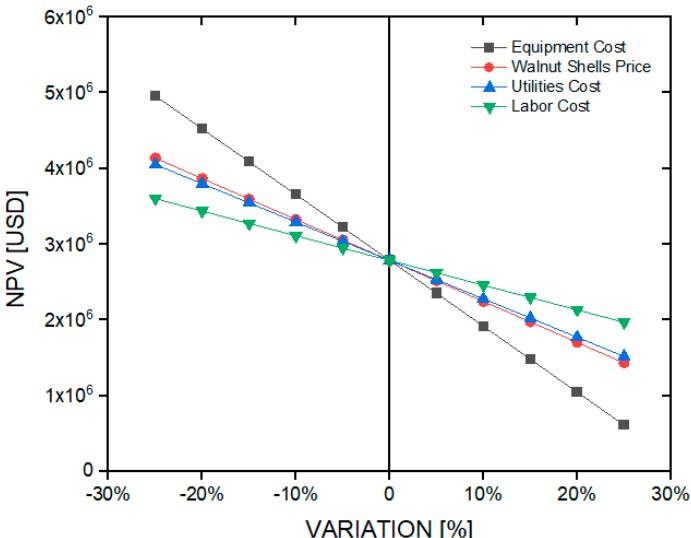

**Figure 4.** Effect by variation in ±25% of the cost of equipment, the purchase price of nutshells, the cost of services, and the cost in operating labor on the NPV considering an investment project of 10 years of production.

When the costs of the basic services were modified, the NPV had a ±46% variation lower than in the previous cases. Thus, the NPV reached a minimum value of USD 1,518,102 when the cost of basic services increased by 25%, obtaining an IRR and ROI value of 16% and 21%, respectively. Finally, the NPV had an ±29% variation when the cost of operational labor was modified for the base case (USD 280,800). It was the smallest variation within the cases studied in this particular item. In this case, the NPV minimum was USD 1,969,277, when the operational labor cost increased by 25%, obtaining an 18% IRR and an ROI of 22%.

Figure 5 shows the variation in net present value for the ±25% change in the volume of raw material processing (nutshells) and in the sales price of the finished product (activated carbon), which for the base case under consideration were 10 kton/year and 2.75 USD/kg, respectively. The NPV varied by up to ±178% when the processing volume of the raw material was modified to the base case. When the processing volume decreased by 25%, a minimum NPV with a negative value of USD 2,170,121 was obtained.

The NPV had a ±234% variation when the sales price of the finished product was modified to the base case. When this sales price decreased by 25%, a minimum NPV with a negative value of USD 3,731,774 was obtained. Consequently, the final product's sales price is the parameter with the highest sensitivity of the series studied, as it delivers the most upper range of variation in the net present value of the net cash flow of the investment project at 10 years of production.

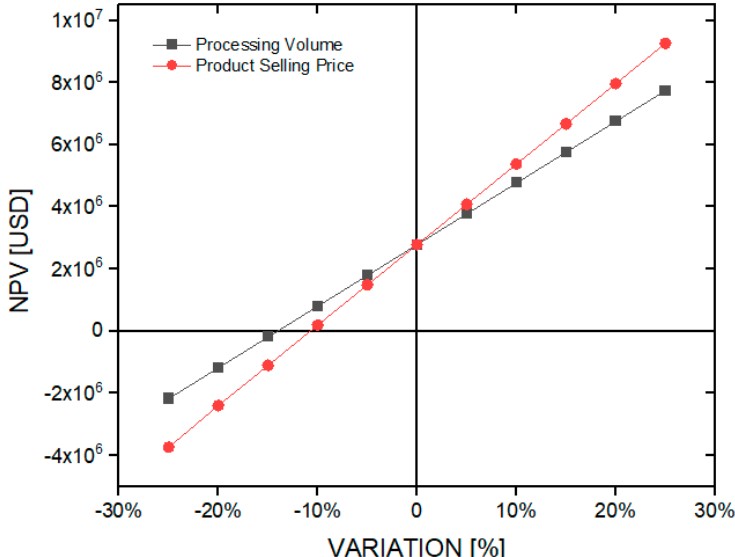

**Figure 5.** Effect on the NPV by variation in ±25% in the volume of processing of raw material (nutshells) and the selling price of the finished product (activated carbon) for the investment project of 10 years of production.

### 4.3. Economic Evaluation (15-Year Production Project)

Similar to the previous case, Figure 6 shows the cumulative discounted cash-flow diagrams at different discount rates to determine the payback period. For a zero discount rate, the payback period was after the fourth year of production, giving a net present value of USD 14,189,402. For a 10% discount rate, the payback period was after the fifth year of production, giving a net present value of USD 4,519,482. For a discount rate of 20%, the payback period was after the ninth year of production, giving a net present value of USD 686,744. In all cases the return on investment was 24%. Figure 7 shows the variation curve of the net present value at different discount rates. From this figure, the cost of the internal rate of return (IRR) can be graphically estimated at 23%. Therefore, the project is feasible for this particular condition, as discussed above.

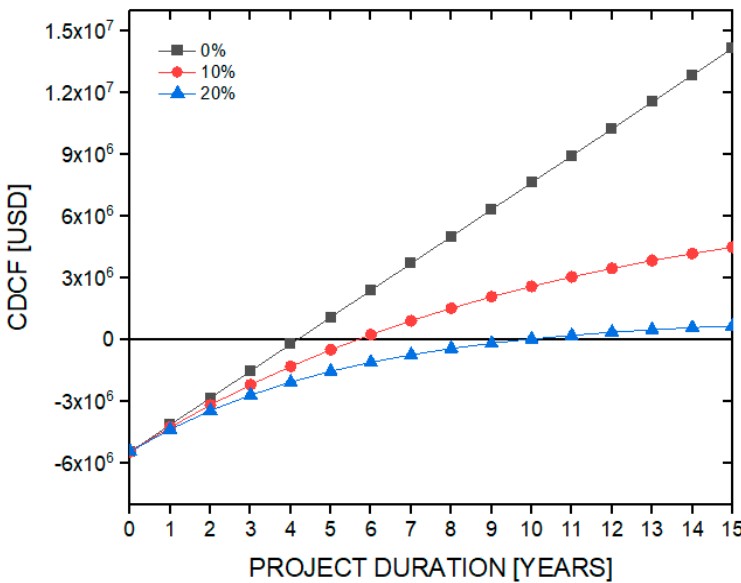

**Figure 6.** Cumulative discounted cash-flow diagrams (CDCF) at different discount rates of 0%, 10%, and 20% for the 15-year production investment project.

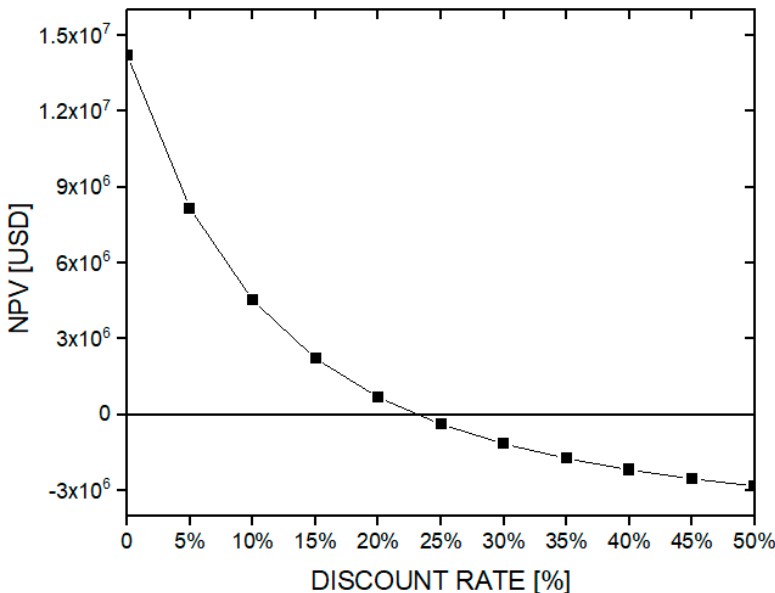

**Figure 7.** Net present value (NPV) of the investment project, at different discount rates for the 15-year production investment project.

*4.4. Sensitivity Analysis (Base Case of 15 Years of Production)*

Figure 8 shows the change in the NPV of the project cash flow at 15 years of production as a base case (using a tax rate of 27% per year and an LDR of 10% per year) due to the variation in ±25% in the cost of the equipment delivered (including auxiliary equipment), the price of the raw material (nutshells), the cost of basic services, and the cost of operational labor. This analysis is similar to the case seen above, which involves a 10-year production project. The NPV varied by up to ±54% when the cost of the equipment delivered was modified from the base case (USD 1,063,044). When the delivered equipment cost was increased by 25%, a minimum NPV was obtained of USD 2,096,311, along with an IRR of 15% and an ROI of 17%.

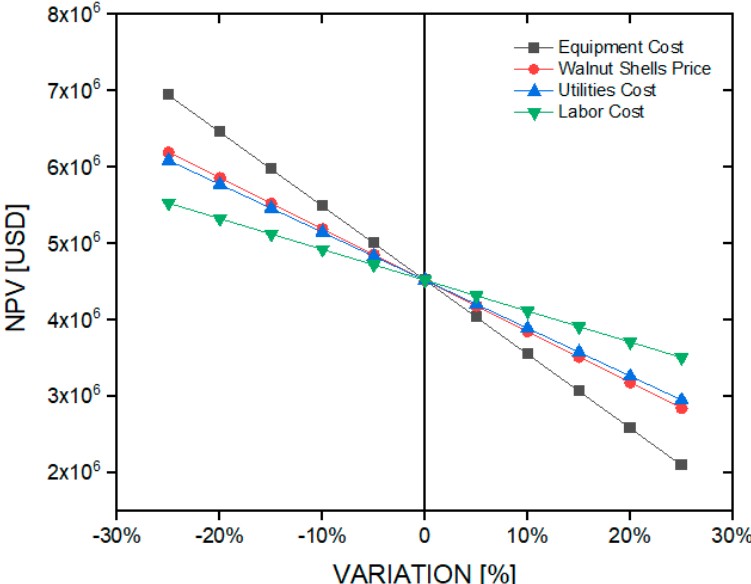

**Figure 8.** Effect by variation in ±25% of the cost of equipment, the purchase price of nutshells, the cost of services, and the cost in operating labor on the NPV considering an investment project of 15 years of production.

The NPV had a maximum variation of ±37% when the price of the nutshells' raw material was modified for the base case (100 USD/ton), this variation being lower than the previous case. When the price of the raw material increased by 25%, a minimum NPV for this case of UDS 2,844,866 was obtained, along with an IRR of 18% and an ROI of 20%. When the costs of the basic services were modified, the NPV had a ±35% variation lower than in the previous cases. Thus, NPV reached a minimum value of USD 2,950,474 for a +25% variation in the cost of basic services, obtaining a 19% IRR and a 20% ROI. Finally, the NPV had a maximum difference of ±22% when the cost of operational labor was modified to the base case (USD 280,800), the lowest variation being within the cases studied in this particular item. In this case, a minimum NPV of USD 3,508,962, an IRR of 20%, and an ROI of 22% were obtained if the operational labor cost is increased by 25%.

Figure 9 shows the change in net present value for the ±25% change in the volume of processing of the raw material and the selling price of the finished product for their base cases discussed above. The NPV had a variation of up to ±136% when the processing volume of the raw material was changed from the base case. When the processing volume decreased by 25%, a minimum NPV with a negative value of USD 1,615,009 was obtained. On the other hand, the NPV had a ±177% variation when the sales price of the finished product was modified for the base case. When this sales price decreased by 25%, a minimum NPV with a negative value of USD 3,496,851 was obtained. Consequently, the sales price of the finished product is the most sensitive parameter of the series studied for the 15-year production investment project, in a similar way to the 10-year operation project.

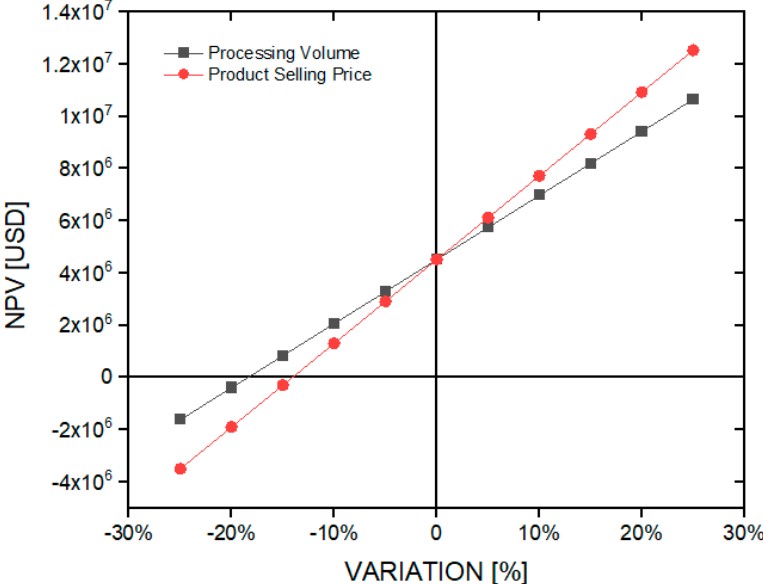

**Figure 9.** Effect on the NPV by variation in ±25% in the volume of processing of raw material (nutshells) and the selling price of the finished product (activated carbon) for the investment project of 15 years of production.

## 5. Conclusions

Based on the economic analysis developed, it is estimated that the generation of activated carbon from industrial waste nutshells by physical methods through the use of water vapor as an activating agent is economically profitable under the particular characteristics of the cases studied in this work. The activated carbon from nutshells would have a total cost of approximately USD 2.15 per kilogram of final product. In addition, the total manufacturing cost reached a value of USD 1.82 per kilogram of final product. For the calculation of these values, a non-zero cost was considered for the purchase of the raw material. This consideration makes a difference with other published studies that consider a marginal cost or do not consider the costs of purchasing raw materials. The estimated internal rate of

return was 21% and 23% for 10 and 15 years of operating time, respectively. Such results are higher than the typical minimum discount rate of 10%, which is generally used by companies to evaluate investment projects in chemical plants, thus making the investment project viable. It was estimated that for the projected case of 10 years of operation (tax rate of 27% and LDR of 10% per year), the return on investment reaches 25%, and that for the planned case of 15 years of service (tax rate of 27% and LDR of 10% per year) the return on investment reaches 24%. In both cases, the payment period was made after the fifth year of production. The analysis of various sensitivity factors showed the limits to ensure the profitability of the project. Among these factors, the selling price of the finished product is the most sensitive parameter. On the other hand, one of the most important contributions of this work is to try to reduce the degree of uncertainty in the estimation of production costs and in the analysis of profitability indicators to provide a better approximation of the real costs involved in the economic analysis of an activated carbon production plant and provide a greater amount of information for potential industrial investors. In summary, this study can support the academic, research and financial analysis of investment projects and provide valuable information to industrial investors who could identify a good return on their investment capital.

**Author Contributions:** Conceptualization, M.L. and J.S.; Methodology, M.L. and J.S.; Validation, All Authors; Formal Analysis, M.L. and J.S.; Investigation, M.L. and J.S.; Resources, M.L. and J.S.; Data Curation, All Authors; Writing—Original Draft Preparation, M.L., J.S. and S.C.; Writing—Review & Editing, All Authors; Visualization, All Authors; Supervision, M.L. and J.S.; Project Administration, M.L. and J.S. All authors have read and agreed to the published version of the manuscript.

**Funding:** This research received no external funding.

**Conflicts of Interest:** The authors declare no conflict of interest.

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
