# Peer review of "Design, Cost Estimation and Sensitivity Analysis for a Production Process of Activated Carbon from Waste Nutshells by Physical Activation"

_processes, doi:10.3390/pr8080945_

Round 1
Reviewer 1 Report
When reviewing scientific papers for publication, I usually start with a general overview in terms of a structure, abstract, literature review, methodology, findings of the research, discussion, conclusions, as well as limitations of the study.
The reviewed paper entitled “Design, cost estimation, and sensitivity analysis, for a productive process of activated carbon from waste nutshells by physical activation” is generally structured in a proper way. There are, however no sections ‘limitations of the study’, and ’future directions of the research”. These sections should be added too, given this is a research paper.
The literature review is average but is founded in the existing literature of the topic. Generally I claim that Author (s) provide good theoretical foundations for the analysis using appropriate references. I would, however, recommend to add some references devoted to the latest literature associated with the topic in question (including Web of Science and Scopus papers).
One should emphasize that the whole paper is very coherent and particular sub-parts fit together.
Additionally, one can see a smooth movement from one point to the another (end of deliberations in one sub-chapter creates also a beginning of a discussion in the next one).
Discussion on the results achieved is presented afterwards. Formulated conclusions prove fulfilment of the stated objective, and provide a good summary evaluation of author’s findings.
Generally my opinion is very positive. Though I have some remarks which may improve the paper , I recommend to accept the paper for publishing after minor revision.
Author Response
"Please see the attachment."

Reviewer 2 Report
I have made carefully revision of the manuscript submitted for publication entitled “Design, cost estimation, and sensitivity analysis, for a productive process of activated carbon from waste nutshells by physical activation”. In my opinion, it is not suitable to be published in Processes
Here are some comments about the manuscript:
- The quality of equations seems to be poor
- References 19 and 20 are not scientific quality and those should be replaced with scientific journal
- Introduction part seems to be irrelevant to the manuscript. In introduction authors discussed e.g copper removal that was not discussed during manuscript at all
- The novelty point should be described. Authors said that there already exists some report on the analysis of production costs of activated carbon production plants. What is the novelty point of this manuscript?
- Where authors obtained the numbers e.g for Table 1? This was unclear for reader.
- Authors have used e.g prices of diesel oil and cooling water without any references? This is weird, typically e.g monthly price or daily price are used and added as reference. Same issue e.g price of employee per hours, is this mean value or what?
- If the audience is industrial investors, is Processes the correct place to publish this?
Author Response
"Please see the attachment."

Reviewer 3 Report
The manuscript Processes-870682 entitled “Design, cost estimation, and sensitivity analysis, for a productive process of activated carbon from waste nutshells by physical activation” presents new economic approaches on the implementation of an industrial plant for physically activated carbon from nutshells.
I have carefully read the manuscript and find the research work relevant, fitting the scope of the journal and likely to be of interest to a broad readership. The manuscript, however, needs some major revisions to be considered as publishable in the journal MDPI Processes. In order to improve the quality of the manuscript, authors may take the following recommendations and comments into consideration:
- What is the scientific contribution of this manuscript? Provide stronger arguments, especially in the Discussion and Conclusion parts. Otherwise, this manuscript looks more like a classical economic analysis / standard report for a case study rather than a research article.
- In the manuscript title “productive process” can be substituted with “production process” of activated carbon.
- Line 12: Better replace “industrial production plant of activated carbon” with “industrial production plant FOR activated carbon”.
- Line 17: Delete “by” in the statement “processing varied by up to 25%”.
- Lines 28-34, Introduction: Activated carbon is indeed suitable for the adsorption of metals from wastewater. Nevertheless, in municipal wastewater treatment a much more important application of activated carbon is for the removal of organic compounds, especially trace organic micropollutant such as pharmaceutical residues and other persistent organic pollutants which cannot be removed through the conventional wastewater treatment process. There are plenty of full‑scale operational plants with an integrated activated carbon final treatment (polishing) stage.
- Line 40: Describe very briefly what is the role of the chemical agents used for the impregnation of the raw material, i.e. what is the purpose of the impregnation for the chemical activation.
- Line 51: Substitute “Exists” with “There are”.
- Line 66: The word “study” repeats in the same sentence. Maybe replace “economic study” with “economic investigation”.
- Line 84: Better “Production process” instead of “Productive process”. This is a general comment which applies to the whole manuscript.
- Line 99: Explain briefly why the considered variation (±25%) in production yield is so high.
- Line 169: Fix the typo: “Cost estimation”.
- Line 308: Add the word “years” after “10 and 15”.
- Line 316: The concluding remark that the activated carbon from nutshells would have a total cost of approx. US$ 2.15 per kilogram of final product can be supported with more detailed calculations in the earlier sections of the manuscript.
Author Response
"Please see the attachment."

Round 2
Reviewer 2 Report
Authors have done great work with improving the quality of the paper. In my opinion, it can be published in present form.
Reviewer 3 Report
The authors have addressed adequately all reviewers' comments and the necessary revisions have been made based on the reviewers' recmmendations. In this current form the manuscript may be accepted for publication in MDPI Processes.